# Multi-Agent Modeling and Jamming-Aware Routing Protocols for Movable-Jammer-Affected WSNs

**DOI:** 10.3390/s23083846

**Published:** 2023-04-09

**Authors:** Biao Xu, Minyan Lu, Hong Zhang

**Affiliations:** 1The Key Laboratory on Reliability and Environmental Engineering Technology, Beihang University, Beijing 140191, China; xubiaorms@buaa.edu.cn (B.X.);; 2School of Reliability and Systems Engineering, Beihang University, Beijing 140191, China

**Keywords:** wireless sensor networks, movable jammer, jamming-aware, routing protocol

## Abstract

Wireless sensor networks (WSNs) are widely used in various fields, and the reliability and performance of WSNs are critical for their applications. However, WSNs are vulnerable to jamming attacks, and the impact of movable jammers on WSNs’ reliability and performance remains largely unexplored. This study aims to investigate the impact of movable jammers on WSNs and propose a comprehensive approach for modeling jammer-affected WSNs, comprising four parts. Firstly, agent-based modeling of sensor nodes, base stations, and jammers has been proposed. Secondly, a jamming-aware routing protocol (JRP) has been proposed to enable sensor nodes to weigh depth and jamming values when selecting relay nodes, thereby bypassing areas affected by jamming. The third and fourth parts involve simulation processes and parameter design for simulations. The simulation results show that the mobility of the jammer significantly affects WSNs’ reliability and performance, and JRP effectively bypasses jammed areas and maintains network connectivity. Furthermore, the number and deployment location of jammers has a significant impact on WSNs’ reliability and performance. These findings provide insights into the design of reliable and efficient WSNs under jamming attacks.

## 1. Introduction

Wireless sensor networks (WSNs) are networks of sensor devices that can monitor and record physical or environmental conditions, such as temperature, humidity, sound, etc., and communicate wirelessly with a central location [1]. WSNs have many applications and benefits, such as data center management [2], environmental monitoring [3], health care [4,5], smart homes [6,7], smart military [8], etc. It is evident that many of these application scenarios are safety-critical or mission-critical, where the failure of a sensor node or the corruption of data can lead to significant consequences, such as damage to the environment, loss of assets, or even loss of human lives. Therefore, ensuring the reliability of WSNs is of utmost importance.

However, WSNs frequently encounter numerous security challenges and vulnerabilities as a result of their low level of security, limited resources, shared medium, and dynamic topology [9], and these threats naturally reduce the reliability of WSNs. Jamming attacks are among these vulnerabilities. Jamming is a form of denial-of-service attack that disrupts wireless communication between sensor nodes by emitting noise or signals on the same frequency band [10]. This type of attack can reduce network performance, decrease throughput, increase energy consumption, disrupt routing protocols, and so on [10]. Several techniques have been proposed to prevent or mitigate jamming attacks on WSNs, such as encryption schemes [10], intrusion detection systems [10,11], agent-based models [12,13], and evolutionary game theory [14,15]. Additionally, an interesting fact we have noticed is the lack of research on movable jammers, as also highlighted in several surveys on jammers [16,17]. This refers to situations where the position of a jammer device can change over time. In reality, for war or malicious damage scenarios, movable jammers pose a significant problem and can amplify the destructive power of attacks, while also making it challenging for defenders to detect and counter their presence. Furthermore, it is imperative to investigate the design of a corresponding defensive confrontation mechanism to counter the threat of a mobile jammer.

To model the reliability of WSNs, several approaches have been proposed in the literature. These approaches include statistical models, probabilistic models, and simulation-based models. These models aim to capture the various factors that affect the reliability of WSNs, including the sensor node characteristics, network topology, and communication protocols. Considering the ad hoc network characteristics of sensor nodes in WSNs and the mobility of jammer devices, the multi-agent modeling method is a suitable approach for reliability modeling.

Therefore, to analyze the impact of movable jammers on the reliability and performance of WSNs, this paper proposes a multi-agent approach that includes various agents such as sensor nodes, base stations, and jammers. The jammers utilize constant jamming, are mobile, and have various configurations. Additionally, the study develops a jamming-aware routing protocol, called JRP, to enhance the resilience of WSNs against jamming. JRP utilizes two independent factors, depth and jamming. The jamming ensures that the constructed routing path avoids the jammer’s affected area, while the depth guarantees that the sensor node selects the next information relay node that is closer to the base station. The final route construction considers the weightage of these two factors. The main contributions of this research include:Investigation of movable jammers: Our study is pioneering in its investigation of movable jammers as no previous research has explored this area. We present the first investigation of the impact of movable jammers on WSN reliability and performance utilizing a multi-agent approach.High routing reliability under jamming: The proposed jamming factor, utilizing jamming strength data acquired by sensor nodes, guides the selection of information relay nodes, avoiding those within affected areas.Adaptability and robustness: The addition of a local maintenance mechanism to JRP enhances network adaptability, while the global maintenance mechanism ensures network robustness. The integration of both mechanisms improves WSN reliability in the presence of jamming.

The paper’s remaining sections are structured as follows: Section 2 describes the research background and related work. In Section 3, the authors illustrate the cross-domain multi-agent modeling approach and present the JRP design, the auxiliary maintenance mechanism, and simulation parameter configurations. Section 4 discusses the reliability and performance impact of various jamming scenarios on WSNs. Finally, conclusions and future prospects are drawn in Section 5.

## 2. Background and Related Work

### 2.1. WSN and Its Architecture

WSNs [18,19] are a type of distributed network system that consists of a large number of small, low-cost and low-power devices called sensor nodes, which are equipped with sensors, processing units, and wireless communication capabilities. These nodes are typically deployed in a remote or harsh environment to monitor physical or environmental conditions, such as temperature, humidity, light, sound, vibration, and motion. However, their reliability and robustness in adverse conditions remain a major challenge, which has led to extensive research on WSN reliability modeling and analysis.

WSN architecture can be divided into three layers [20,21]: the physical layer, the network layer, and the application layer. The physical layer [20,21] is responsible for data acquisition and transmission and includes the sensor nodes, the data transmission medium, and the physical interface. The network layer [20,21] provides the routing and data aggregation functionalities and consists of the communication protocols and the routing algorithms. The application layer [20,21] defines the tasks and objectives of the WSN and includes the data processing and analysis algorithms, as well as the user interfaces.

### 2.2. Jamming Attacks in WSNs

Jamming attacks, refer to the intentional interference of the wireless communication channel in a WSN, which can disrupt the transmission of data between the nodes. Jamming attack is a major security threat to WSNs, which can significantly affect their reliability and performance. A number of studies have investigated the impact of jamming attacks on WSNs and proposed various solutions to mitigate their effects. Several studies have proposed different types of jamming attacks, such as constant jamming [21], deceptive jamming [22], random jamming [20], and reactive jamming [23], and evaluated their impact on WSN reliability.

For instance, in [24,25], the authors focus on the attacker’s strategy. In the former paper, the authors optimize the best relay amplification matrix of WSNs and concluded that the network performance is influenced by the geometric shape of the nodes. In the latter paper, the authors consider the wireless relay network under a jamming attack, where the node’s receiving power is limited. The author proposes a strategy to calculate the diagonal relay amplification matrix to minimize the mean square error between the transmitted signal and the received signal.

Other studies analyze the detection and location of the jammer from the defender’s perspective, providing the possibility of actively eliminating the jammer. Wood et al. [26] introduced the JAM protocol, which detects and maps jammed areas in wireless sensor networks, treating the disturbed area as a single entity, instead of a set of disconnected links and congested nodes. The protocol uses unperturbed nodes around the disturbed node to collect and disseminate information about the affected area and employs a gradient-based algorithm to calculate the boundary of the disturbed region. Liu et al. [27] proposed a method to localize single or multiple jammers by exploiting the neighbor changes caused by jamming. The authors analyzed the impact of different types of interference on the neighbor discovery process and neighbor table maintenance process and proposed an algorithm to estimate the jammer’s position based on the neighbor change frequency and directional information. They also designed a distributed cooperation mechanism that allows nodes in the network to share and fuse their respective observed neighbor change information, thereby improving the positioning accuracy. Liu et al. [28] systematically analyzed the impact of multiple overlapping or non-overlapping interference regions on network topology and connectivity, proposing a framework that can divide the network into different clusters and estimate multiple jammer positions. The framework utilizes two techniques: partitioning clusters based on minimum spanning trees (MST) and estimating locations based on maximum likelihood (ML). The framework can efficiently handle multiple overlapping or non-overlapping interference regions with low communication overhead and computational complexity.

Another form of defense is to improve the network’s robustness. Wang et al. [29] proposed a cluster-based cooperative jamming (CB-CJ) scheme for improving security in wireless multi-hop networks. This scheme uses friendly nodes in the network as jammers to send artificial interference signals to enemy eavesdroppers, thereby reducing their signal-to-noise ratio and improving the confidentiality rate between legitimate communicators. This scheme adopts a method of selecting the optimal relay and jammer on demand and considers different security requirements and power constraints. Wood et al. [30] analyzed various Denial-of-Service (DoS) attacks against sensor networks and their defense mechanisms. They pointed out that sensor networks are more vulnerable to DoS attacks than traditional networks due to their resource-constrained, distributed, and self-organizing characteristics. Using two examples of efficient sensor network protocols, they showed that designing with security in mind is the best way to ensure network availability. They also discussed some possible security mechanisms, such as authentication, encryption, privacy protection, intrusion detection, etc.

There is also a significant category of jamming research that employs game theory to optimize the action strategy of one or more parties. Clark et al. [31] studied an active defense mechanism against jamming attacks in multi-hop relay networks, utilizing one or more network sources to introduce a spoofed network flow on disjoint routing paths, confusing the interferer and making it impossible to distinguish between real and fake communication flows. They analyzed the impact of deceptive routing on network performance and security and proposed a game theory-based approach to optimize deceptive routing strategies. Zhu et al. [32] have proposed a game theory-based approach to analyze and design solutions for eavesdropping and jamming problems in next-generation wireless networks. The authors have considered three types of participants in the game—legitimate users, eavesdroppers, and interferers, and established game models in various scenarios, such as static games, dynamic games, and repeated games. The study has focused on the strategic interactions and equilibrium outcomes among the players and has proposed algorithms and mechanisms to enhance network security and efficiency. 

The impact of jamming on WSNs and its mitigation strategies have been extensively studied, and numerous review articles have been published [16,17,33,34]. These articles not only categorize the different types of jamming attacks but also explore anti-jamming techniques at various levels of WSN. However, surprisingly, there has been no research on the effects of movable jammers. Therefore, one of the primary objectives of this study is to address this gap in the literature.

### 2.3. Agent-Based Models

Agent-based models are computer simulations used to study the interactions between people, things, places, and time [35]. They are often used to understand complex phenomena that emerge from individual behaviors and interactions.

Agent-based models have been applied to study jamming attacks in various network environments. For example, some researchers have used agent-based models to analyze how jamming attacks affect network traffic [36], how negotiation mechanisms can defend against jamming attacks in smart grid power markets [37], or how machine learning techniques can detect jamming attacks in 5G networks [38].

Multi-agent systems have also become a prevalent approach in WSN research due to their compatibility with the distributed nature of WSNs. For instance, Wu et al. [39] described a method for designing a multi-agent system for structural health monitoring utilizing a wireless sensor network, along with a specific platform dedicated to the network. Sardouk et al. [40] proposed a strategy for optimizing WSN performance using multi-agent systems that consider various parameters, including energy, distance, and delay.

When considering research solely focused on jamming in multi-agent WSNs, only two studies [41,42] utilizing game theory as an analytical method were identified. Zhang et al. [41] propose a novel anti-jamming method based on multi-agent reinforcement learning for wireless networks under jamming attacks. The method uses a partially observable Markov decision process to model the interactions between legitimate nodes and jamming nodes and applies a decentralized Q-learning algorithm to learn the optimal anti-jamming strategy. The paper shows that the proposed method can achieve better performance than existing methods in terms of throughput, convergence speed and robustness. Chen et al. [42] propose a novel game-theoretic framework based on multi-agent deep reinforcement learning for anti-jamming in wireless networks. The paper uses a generative adversarial network to generate realistic jamming signals and a policy gradient algorithm to learn the optimal anti-jamming strategy. The paper shows that the proposed method can achieve better performance than existing methods in terms of throughput and robustness.

## 3. Methodology

### 3.1. Multi-Agent Modeling Approach

To utilize the multi-agent modeling approach for analyzing the effect of movable jammers on WSN reliability, it is crucial to initially model the various agents that exist in the scenario. This paper presents a multi-agent modeling method that includes three distinct agent types: sensor agent, base station agent, and jammer agent.

#### 3.1.1. Sensor Agent

Wireless sensor nodes serve as network nodes with ad-hoc network functions and as communication relays. The ad-hoc network function depends on the routing protocol processed by the computing component, while the communication relay function relies on the communication component that receives and sends radio signals. Additionally, the power supply component is essential for these components to function. Figure 1 illustrates the structure of a wireless sensor node.

A sensor node typically comprises four main components, namely the sensing component, computing component, communication component, and battery component, as depicted in Figure 1.

As this paper focuses on the impact of jamming on WSN reliability, it is important to abstract sensor nodes into agents by considering their activities and state transitions in the network layer.

The primary function of sensors is to collect environmental data, which are then uploaded and concentrated to the base station via the ad hoc network. To achieve the self-organizing network function, routing protocols must be deployed to help sensor nodes select successor nodes closer to the base station. In the context of jamming, routing protocols should also enable sensor nodes to avoid affected areas, select an alternative successor node when the current node is interfered with, and maintain normal data transmission. Section 3.2 details the design of the routing protocol.

Under strong jamming, the signal-to-noise ratio of sensor nodes decreases, causing communication failure, while weak jamming still allows nodes to act as information relay nodes. In both cases, a sensor node can broadcast information about its jammed state and notify other unaffected nodes to consider other successor nodes, reducing the probability of relay node disconnection from the base station.

#### 3.1.2. Base Station Agent

In a WSN, the base station is responsible for collecting and summarizing data from the sensor nodes, as well as performing data fusion, analysis, storage, and forwarding. To enable distributed sensor nodes without global information to have the function of ad hoc networking and transfer data to the base station, a routing protocol was designed (see Section 3.2). The base station performs two functions: firstly, it performs depth broadcast (see Figure 2) during network initialization and assigns initial values to the depth of all sensor nodes, with the help of node-by-node forwarding from the sensor node; secondly, it performs depth broadcast again during global maintenance and updates depth assignments for all sensor nodes using sensor node-by-node forwarding.

As depicted in Figure 2, the central large solid blue circle portrays the base station, while the small black hollow circle represents the sensor node. The dotted circle denotes the respective communication ranges of the base station and the sensor node, and the blue arrow indicates the propagation direction of depth information. It is evident from Figure 2 that the communication range of the base station exceeds that of the regular sensor nodes. Consequently, sensor nodes within the communication range of the base station can directly communicate with the base station, and their depth value is set to 1. Subsequently, nodes with a depth value of 1 relay the depth broadcasting, assigning a depth of 2 to nodes that have not received a depth value within their communication range. This process continues, and the depth broadcast emanates from the base station, spreading outwards until all sensor nodes have been assigned a depth value. The depth value of the base station is 0, signifying the shallowest depth. As the distance from the base station increases, the depth value increases, indicating greater depth.

Considering that the base station has stronger computing power and a larger antenna area than ordinary sensor nodes, its information transmission range should also be larger. Sensor nodes farther away from the base station can also directly establish a connection with the base station. Furthermore, while the jammer cannot disable the communication function of the base station, it can still disrupt data collection by destroying the communication function of the sensor nodes around the base station and reducing the number of relay nodes.

#### 3.1.3. Jammer Agent

A jammer is a transmitter that disrupts electronic signals, such as radio and radar signals. In this paper, we focus on the impact of movable jammers on the reliability of WSNs. The jammer considered in this scenario is an active jammer that transmits jamming signals to disrupt communication between surrounding sensor nodes and base stations.

One of the defining properties of jammer agents is their mobility. Therefore, we designed two mobility modes for jammer agents: fixed and random mobility. The interference signal sent by the jammer weakens with distance, so closer sensor nodes will have their communication transmission capabilities completely destroyed, making it impossible to transmit data to other nodes and base stations. However, nodes that are farther away from the jammer can still maintain their connection to other nodes and base stations. Hence, the range of influence is another critical attribute of jammers. We define two ranges of influence for jammers, one being slightly smaller, affecting only nodes within its coverage, and the other having a broader coverage.

Additionally, the deployment location of the jammer, whether on a UAS (unattended aerial system) or a ground vehicle, also affects the coverage of its jamming signal. Therefore, we designed two groups of influence ranges for jammers corresponding to UAS and ground-based. However, the coverage radius of a jamming device with the same power is significantly different when mounted on a UAS and a ground vehicle. For instance [43], the coverage radius when mounted on a UAS is 75 km, while on a ground vehicle, it is only 7.1 km.

Finally, the number of jammers has a significant impact. A single jammer can destroy the information relay function of a sensor node. However, our routing protocol allows the message transmission line to bypass the jammed area and re-establish contact with the base station. When multiple jammers work in tandem, they can cause the message transmission line to fail to bypass the jammed area, making it impossible to upload sensing data to the base station.

### 3.2. Jamming-Aware Routing Protocol (JRP)

#### 3.2.1. Routing Factors Design

WSN deployment is mostly conducted in unattended environments. This makes the data transmission process of sensor nodes vulnerable to malicious factors. Therefore, the routing protocol should avoid information transmission paths through jammed areas to reduce the impact of the jammer on the routing performance. This routing idea has already been applied to robot navigation, where the robot needs to avoid high-risk areas (e.g., areas with jammers). Regarding sensor nodes, they can collect electrical signals of their location through antennas, and also obtain jamming data of their adjacent areas through message exchange between adjacent nodes.

As we know, the intensity of signals decreases with an increase in transmission distance. This is applicable to both communication transmission and communication interference. Thus, focusing on the jammer, an interference intensity (we call it jamming) is formed. The intensity that decreases gradually from the inside to the outside is a high-risk area for sensor communication. Therefore, routing protocols should avoid information transmission paths that pass through such areas.

Of course, in order to provide the most fundamental routing function, each sensor node should also possess the depth information of itself and its surrounding sensors, which refers to the number of hops along the feasible shortest path to reach the base station. Hence, the base station is designated with a depth of 0, and the depth of a node increases as its distance from the base station increases. By selecting the node with the smaller depth as the successor node each time, it can be ensured that the data will eventually reach the base station.

The JRP uses the sensor node’s depth and jamming values as input, which can be obtained through environment sensing, and message exchange with neighboring nodes. Since JRP follows the distributed routing paradigm, its output provides the ID of the next-hop node of the message to guide it to the base station through multiple hops.

#### 3.2.2. Path Discovery

In the initialization phase, an initial value is assigned to the depth and jamming of each sensor node. The depth of the base station is set to 0, and a depth broadcast is initiated by the base station to assign depth values to each sensor node. The sensor nodes update their depth information and relay the depth broadcast based on their comparison of the received depth value. Similarly, the jamming value of each sensor node is set to 1 initially, and the jamming influence of the jammer is applied to update the jamming value of the sensor nodes. In order to determine the target value of each sensor node, the reciprocal of its depth and jamming values are summed and divided by 2. This target value is then broadcast to surrounding nodes to guide their selection of a successor node. The formula used to calculate the target value is given by
(1)$$T_{i}=\frac{\frac{1}{D_{i}}+J_{i}}{2}$$

Among them, $T_{i}$ is the target value of sensor node i, which is used to guide the sensor node to select a successor node, $D_{i}$ is the depth value of sensor node i, and $J_{i}$ is the jamming value of sensor node i.

During the path discovery stage, each sensor node selects the node with the highest target value as its successor node. The node then repeats this process until it reaches the base station. This process results in the formation of multiple paths from the sensor nodes to the base station, which ensures the reliable transmission of data. For example, Figure 3 illustrates the base station using a hexagon, while the sensor node is represented by a circle. The arrow denotes the actual direction of information transmission, whereas the dashed line signifies the potential communication connection. Node 6 selects either node 4 or node 5, depending on which has the higher target value, as its successor node to transmit its sensing data to the base station. This process is repeated until the base station is reached, resulting in two paths 6→4→3→1→base station and 5→2→base station that cover all the current sensor nodes.

In summary, during the initialization phase, the depth and jamming values of each sensor node are assigned, and the target value is calculated and broadcasted to surrounding nodes. During the path discovery phase, each sensor node selects the node with the highest target value as its successor node to transmit its data to the base station.

#### 3.2.3. Routing Maintenance

Due to the presence of the jammer, and particularly considering its mobility, the jamming value of a sensor node may change over time, leading to a possible failure of the node’s successor node due to jamming. To maintain connectivity in the WSN, it is necessary to design a reasonable JRP maintenance mechanism, for which we propose two approaches.

The first is local maintenance, in which a sensor node recalculates its target value and broadcasts a target value update when it perceives a change in its interference intensity that exceeds a threshold. Sensor nodes receiving the broadcast will recompare their own target values and select the node with the highest target value as the successor.

The second approach is global maintenance, in which, after the WSN operates for a period of time, the base station initiates a depth broadcast to update the depth values of the sensor nodes. This is necessary because the presence of a jammer and the local maintenance mechanism can disrupt the propagation of messages, leading to an increase in the number of hops required for sensor nodes to deliver messages to the base station. The global maintenance is similar to the depth broadcast in the initialization phase, but some sensor nodes may not receive the depth broadcast due to the jammer, which can affect the global update of the depth value.

### 3.3. Simulation Process

Our simulation process consists of the following nine steps (see Figure 4):

The first step is to deploy a WSN within a specified area. We randomly place a certain number of sensor nodes in a square scene and deploy a base station at the center of the scene.

The second step is to initialize the network depth and jamming. The depth broadcast is initiated by the base station with a depth value of 0, and the sensor nodes receive the broadcast information and relay it. Finally, the initial value of the depth of all sensor nodes is assigned, and the initial default jamming value of all sensor nodes is set to 1.

The third step is to calculate the target value and broadcast it locally. All sensor nodes calculate their own target value based on their depth and jamming value and then initiate a local broadcast to inform other sensor nodes within their surrounding communication range about their target value. The target value of the base station is set to 100, ensuring that any sensor node that can directly communicate with the base station will connect with the base station without selecting other sensor nodes as relays.

The fourth step is to initialize the network. All sensor nodes select the node with the largest target value within their communication range as their successor node, enabling all sensor nodes to connect with the base station and complete the ad hoc network.

In the fifth step, the jammer enters the area, and the sensor nodes update the target value. One or more jammers are deployed randomly in the scene, and they affect the jamming values of sensor nodes within a certain range. The affected sensor nodes recalculate their target value and initiate a local broadcast to inform the surrounding nodes that their target value has been updated, indicating that they are affected by jamming.

The sixth step is local maintenance. After receiving the target value update message sent by other sensor nodes, the sensor node reselects the node with the largest target value as their successor node within their surrounding communication range to avoid the jamming affected area.

The seventh step involves the jammer moving to a new location and affecting sensor nodes within the surrounding range, updating their jamming values. These interfered nodes also need to update their own target values and broadcast them locally. Local maintenance follows closely. The seventh step is similar to the iterative cycle of steps five and six.

The eighth step is global maintenance. After running for a period of time, the structure of the WSN may have changed significantly due to jammers and local maintenance. The communication path between the remote sensor node and the base station may no longer be the shortest path, resulting in an increased delay in information transmission. To reduce the average path length from each sensor node to the base station, the base station periodically initiates the depth broadcast. The sensor node receives the depth broadcast information, relays it, updates its own depth value and the target value, and sends the information to the surrounding sensor nodes within the communication range broadcast. All sensor nodes that receive the depth broadcast information must reselect the node with the largest target value from the surrounding nodes as their successor node. Note, that at this point, the communication function of the sensor node deeply affected by the jammers is invalid, so it will not update its own depth information or initiate a local broadcast of the target value.

The ninth step involves recording the simulation results; the first to the fourth step. After the initialization phase (from the first step to the fourth step), the WSN starts to run and undergoes jammer entry/movement and local maintenance during each time slice, which is followed by updating the current network metrics and recording the results. Additionally, after every fixed period, global maintenance is performed along with jammer entry/movement and local maintenance. Following global maintenance, the current network metrics are updated and recorded.

### 3.4. Simulation Parameters

We conducted simulation experiments to verify our approach using self-developed simulation code in NetLogo. The simulation environment consisted of a 60 × 60 area, which wraps around horizontally and vertically. The 400 sensor nodes were randomly positioned within the area. The initialized network topology was based on the JRP, and an example of the network topology is shown in Figure 5. As the sensor nodes always select another sensor node with the largest target value within their communication transmission range as the information successor nodes, the network is always a preferential attachment structure. This structure is similar to the network generated by the Barabási–Albert algorithm. Among them, the brown dot represents the sensor node, the green pentagon represents the base station, the blue directed link represents the successor node of the sensor node, and the red bug represents the jammer.

The network was subject to jammers, and the interference in the experiment decreases in intensity as the distance from the jammers increases. The central interference intensity of the jammers in this experiment destroys the communication function of the sensor nodes. However, as the distance increases, the interference intensity decreases, and the soft-failure probability of the sensor nodes decreases accordingly. The focus of this paper is on the influence of movable interference sources on the reliability of WSNs. To simplify the experimental design and speed up the simulation experiment, we set two influence ranges for the jammers. Within the smaller influence range, the interference intensity is high, and the affected sensor nodes cannot maintain the communication transmission function. Within a relatively large influence range, the interference intensity decreases, and the affected sensor nodes can maintain normal operation, but they can perceive that they are being interfered with.

The communication transmission range of all sensor nodes was set to 10, and the base station node was located in the center of the simulation area. All simulation parameters are summarized in Table 1, and each simulation experiment design has 10 independent runs.

## 4. Results and Discussion

### 4.1. Performance Metric

In this study, we focused on evaluating the reliability and performance of WSNs based on two key metrics: maximum connected subgraph and average path length.

The maximum connected subgraph (MCS) refers to the graph consisting of all sensor nodes that can connect to the base station and the edges connecting them. The size of the maximum connectivity graph reflects the robustness of the WSN when it suffers from jamming. Specifically, it represents the number of sensor nodes that can upload sensing information to the base station, despite the presence of jammers.

On the other hand, the average path length (APL) is calculated as the average number of hops required for all sensor nodes that can communicate with the base station to reach it. The higher the number of hops required, the greater the delay in information transmission and the lower the performance of the network. Thus, the average path length metric is a crucial factor in assessing the efficiency and effectiveness of WSNs.

It is important to note a special case when measuring APL where not all sensor nodes are connected to the base station. In such cases, the number of sensor nodes connected to the base station is 0, which would lead to a division by zero problems. To address this issue, we set the APL value to 6 for this special case. This decision was made based on the observation that the jammer moves to a new position every tick, which triggers local maintenance to be performed spontaneously. As a result, the scenario where all nodes are disconnected from the base station typically lasts for only one tick and is significantly improved in the next tick. Moreover, we have noticed in our simulation experiments that the APL value in such cases usually ranges between 2 and 5. Setting the APL value too high in this scenario could mask the actual network performance observed in the simulation.

### 4.2. Simulation Results and Analyze

#### 4.2.1. Simulation 1

In the first simulation, we aimed to analyze how the mobility of the jammers and local maintenance designed for sensor nodes would impact the WSN’s reliability and performance. We designed four parameter combinations for this experiment: fixed jammer with no local maintenance (FF), fixed jammer with local maintenance (FT), movable jammer with no local maintenance (TF), and movable jammer with local maintenance (TT). For the first two cases, as the jammer does not move, it only takes one tick to observe the impact of the parameter design on the experiment. For the latter two cases, since the jammer is movable, it takes a longer time, and its influence range is much larger than before. Therefore, we designed a running time of 5 ticks for the dynamic situation.

The results in Figure 6a indicate that the continuous movement of the jammer affects a wider range of sensor nodes. The use of local maintenance significantly improves the total MCS performance compared to not using it (FT vs. FF, TT vs. TF), resulting in better reliability. In the case of local maintenance, TT outperformed FT, mainly because the jammer moved farther away from the base station in the fifth tick, resulting in fewer affected sensor nodes.

In Figure 6b, the APL performances of FF, FT, and TT are similar, while only TF’s performance is exceptionally good. This may be due to the fact that in this simulation, the jammer was mostly moving around the periphery of the screen, resulting in a large number of peripheral nodes with a larger number of hops being disconnected from the base station, leaving only the inner peripheral nodes with a smaller number of hops. However, it should be noted that MCS is a more critical indicator than APL because the establishment of a connection between sensor nodes and the base station is a prerequisite for discussing transmission delay.

#### 4.2.2. Simulation 2

In the second simulation, we aimed to analyze the impact of global maintenance on the WSN’s reliability and performance. We designed two cases for this experiment: with and without global maintenance. In the case of local maintenance, the sensor node can choose the node with the largest surrounding target value as the successor node. However, the network structure is affected by the jammer and local maintenance, causing changes in the structure over time, which deviates from the optimal path combination. To show this phenomenon, we designed a longer running time of 100 ticks for the simulation experiment investigating global maintenance.

Comparing Figure 7a and Figure 8a, we observed that in the absence of global maintenance, the value of APL fluctuates slightly, and the average value is roughly the same as that of global maintenance. Figure 9b also confirms this finding; the presence or absence of global maintenance did not appreciably impact APL in a substantively meaningful manner. Although the expected global maintenance was aimed at periodically resetting the depth of the sensor nodes to restore the efficient structure of the sensor node ad hoc network to the initial state, this effect did not materialize.

In contrast, when comparing Figure 7b and Figure 8b, it is evident that the value of MCS not only fluctuates more, but the average value is lower without global maintenance than with global maintenance. Figure 9a also supports this claim. Global maintenance has a considerable impact on MCS. Due to the influence of moving jammers, sensor nodes update their selection of successor nodes multiple times, and eventually, many peripheral sensors choose some of the same sensor nodes as relay nodes. In other words, compared to the initialization network, the number of relay nodes responsible for data relay tasks is less. At this point, if the jammer moves and destroys the communication transmission function of these small number of relay nodes, it will result in a large number of peripheral nodes connected to the base station relying on these relay nodes to be disconnected, as shown in Figure 7b, and the value of MCS plummeted.

From the results of simulation experiments, it can be concluded that the primary function of global maintenance is not to maintain the low latency of the network but to sustain the robustness of the network and avoid relying on a small number of relay nodes.

#### 4.2.3. Simulation 3 and Simulation 4

In the third simulation, we aimed to analyze the effect of the number of jammers on the WSN’s reliability and performance. We designed four different cases for this experiment, with 1, 2, 3, and 4 jammers.

The fourth simulation aimed to examine the impact of jammer deployment location (ground versus air) on WSN reliability and performance. To achieve this, we designed new jammer influence range data, with high and low values of 24 and 18, respectively. Electromagnetic wave propagation attenuation is weaker in the air than on the ground, resulting in significantly larger jamming signal coverage for air-based jammers. Research has shown that the coverage radius is approximately 75 for a jammer mounted on a UAS, compared to only 7.1 on a ground vehicle [43], indicating a substantial disparity in jamming performance. To avoid abrupt experimental results, we reduced the interference performance gap between the two cases to a ratio of 3:2 in the experimental design. However, it is important to note that this ratio would be much greater in real-world scenarios.

Since Simulation 4 has only one set of experimental designs, we combined it with Simulation 3 for comparison purposes.

From Figure 10a, it is evident that as the number of jammers increases, the number of sensor nodes capable of establishing a connection with the base station and uploading data gradually decreases, which is in line with expectations. Furthermore, the jammers mounted on the air platform demonstrated far stronger jamming performance than those mounted on the ground platform, and a single air jammer was more destructive than four ground jammers.

Figure 10b shows that although the MCS is gradually decreasing with the increase in the number of jammers, the value of APL remains stable. However, there was one unexpected result with the air jammer. We can attempt to explain this phenomenon through Figure 11a,b.

From Figure 11a, it is evident that the MCS exhibits a platform trend during certain time periods when the air jammer moves far away from the base station. Although the jammer still interferes with the communication transmission of a large number of sensor nodes due to its broad coverage, it is not fatal. In other time periods, the MCS drops suddenly, and during this time, the air jammer moves to the vicinity of the base station. Although the jammer cannot destroy the communication function of the base station in the experimental design, it can disable the communication function of the sensor nodes around the base station, leaving no relay node to provide message relays for other sensor nodes in the periphery.

The time period of the sudden rise of APL in Figure 11b coincides with the time period of the sudden decrease of MCS in Figure 11a, indicating that the air jammer destroys the communication transmission function of the sensor nodes around the base station, making them unable to undertake the task of information relay. The number of relay nodes is significantly reduced, and other sensor nodes can only choose fewer nodes next to the base station as relays, forcing many sensor nodes to take a detour to connect to these relay nodes, resulting in an overall increase in the APL value of the network.

### 4.3. Comparison with Other Protocols

In this section, we compare JRP with two commonly used routing protocols, namely AODV and DSR. AODV solely concentrates on the depth of nodes and lacks a maintenance mechanism, while DSR also concentrates solely on the depth of nodes and has a local maintenance mechanism to bypass failure nodes that arise during network operation.

As illustrated in Figure 12a, it is evident that JRP outperforms the other two routing protocols, with DSR ranking second and AODV coming in last. Both JRP and DSR have established local maintenance mechanisms that allow them to bypass sudden failure nodes and attain a certain degree of recovery, thus restoring normal functions to some extent after the WSN experiences external shocks. Therefore, in theory, the performance of MCS should be superior to that of AODV, and the results also confirm this point. To further compare JRP and DSR, it is necessary to analyze the design concepts behind the routing protocols and observe the simulation experiment process. Our analysis reveals that JRP comprehensively considers depth and interference when selecting relay nodes, and its maintenance mechanism allows it to bypass failed nodes locally and preferentially select sensor nodes far away from jammers as relay nodes during maintenance. In contrast, DSR only considers the depth factor of each node itself. When the relay node fails and disconnects, it simply selects another node with a shallower depth than the relay node. However, the new relay it chooses is likely to be in the vicinity of the failed node. If the jammer moves in this direction, it will fail immediately. In contrast, JRP is more inclined to choose relay nodes that are far away from the jammer, making it less susceptible to disruption caused by subsequent movement of the jammer. AODV, on the other hand, does not configure a local maintenance mechanism and only refreshes the network depth and structure during global maintenance. As a result, the reliability of the WSN continuously declines between two maintenance cycles, leading to much worse overall MCS performance than that of JRP and DSR.

As illustrated in Figure 12b, AODV outperforms JRP and DSR in terms of APL performance, with JRP and DSR performing comparably. This is because AODV only considers the depth of nodes, resulting in the shortest path generation. If the link is interrupted due to jammers, most of the disconnected nodes are peripheral nodes far away from the base station, adversely affecting the APL performance. In contrast, both JRP and DSR are equipped with local maintenance mechanisms that enable them to bypass failed nodes, resulting in “detours” in the paths they generate and preserving the connection of peripheral nodes to the base station, leading to an increase in APL.

Overall, although AODV shows better APL performance than JRP and DSR, for WSN, maintaining as many sensor nodes as possible takes priority. In terms of MCS performance, JRP outperforms the existing common routing protocols. The simulation experiment has thus confirmed the reliability advantage of JRP over other existing routing protocols in the presence of movable jammers. 

### 4.4. Discussion

From the simulation experiments mentioned above, it is evident that the mobility of jammers has a significant impact on the reliability and performance of WSNs. To cope with this issue, we designed local and global maintenance mechanisms with the aim of addressing short-term problems of updating successor nodes of sensor nodes and reducing overall delay increases caused by continuous network structure updates during long-term operations. Local maintenance functions as expected, but global maintenance does not significantly reduce overall message transmission latency. We believe that this is due to the small area of the scene used in the simulation experiment, which results in sensor nodes having a communication range that is too large compared to the size of the scene. Therefore, even with constant updates to the network structure due to jammers and local maintenance, the number of hops from the base station to the selectable successor node of the sensor node remains small. As a result, the effect of reducing the overall network delay is not significant.

But surprisingly, the global maintenance mechanism can alleviate the tendency of sensor nodes to centrally select a small number of relay nodes. Due to the impact of the jammer and local maintenance, some nodes around the base station that can act as relay nodes are no longer selected by other sensor nodes, leading to a concentration of relay tasks on other relay nodes around the base station. If these relay nodes are also affected by the jammer, it can cause a sudden drop in the number of sensor nodes that can communicate with the base station. However, global maintenance can allow those abandoned relay nodes to be re-selected as relay nodes, thereby increasing the number of redundant relay nodes and improving the network’s robustness.

The simulation experiments also revealed that the number and deployment position of jammers have a significant impact on the reliability and performance of WSNs. As the number of jammers increases, the reliability of WSNs gradually decreases, while the communication delay shows no significant difference. Our simulations also demonstrated that a single air jammer is more destructive than four ground jammers. However, it should be noted that the parameter design used in our simulation experiment is far from the actual scene, and if experimental parameters were configured according to actual scene conditions, the results would differ significantly.

Finally, we compared JRP with other existing routing protocols and verified its reliability advantage in WSN in the presence of movable jammers. The advantages of JRP mainly stem from two factors. Firstly, in the event of a jammer causing soft damage to some nodes and leading to their failure, JRP’s local maintenance mechanism enables other nodes to bypass the failed nodes and restore connections with the base station. Secondly, JRP comprehensively considers both depth and jamming when selecting relay nodes, avoiding nodes closer to the jammer to prevent forced disconnection during subsequent movement of the jammer. 

## 5. Conclusions

Wireless Sensor Networks (WSNs) are crucial for mission-critical applications, and their reliability is vital. However, due to energy and cost constraints, WSNs are vulnerable to malicious attacks, which increases the reliability risks. Additionally, in the current research on WSNs being affected by jamming, there is no research on mobile jammers. To fill this gap, we propose a WSN modeling approach based on multi-agent modeling and jamming-aware routing protocol (JRP). The main contributions include: (1) using multi-agent modeling to study the impact of movable jammers on the reliability and performance of WSN for the first time; (2) the proposed routing protocol can guide sensor nodes to choose a jammer-avoiding path, considering the depth of the nodes and the degree of influence by jamming; (3) designing a maintenance mechanism to improve the robustness of the network.

Simulation results show that the mobility of the jammer has a significant impact on the reliability and performance of WSN, requiring the design of local and global maintenance mechanisms for sensor nodes to maintain the normal operation of WSN. Although the global maintenance mechanism does not show the expected contribution to reducing the information transmission delay of WSN, it can effectively reduce the risk of over-concentration of relay nodes and improve the network robustness of WSN. Furthermore, the simulation experiment confirmed the impact of the number of jammers and the height of their deployment on WSNs. Attackers can significantly weaken the reliability and performance of WSNs by increasing the number of jammers or deploying them in the air.

This paper only considered the jammer’s mobility and did not design scenarios or strategies for the jammer’s movement. The jammer mobile deployment strategy of the attacker and the jammer detection strategy of the defender can be discussed based on game theory. In addition, we only considered depth and jamming as the basis for routing decisions. However, due to the energy limitation of sensor nodes in WSN, their residual energy should also play a role in routing decisions, and nodes with heavier relay tasks consume energy faster. Therefore, how to integrate depth, energy, jamming, and other information will be the focus of future research.

## Figures and Tables

**Figure 1 sensors-23-03846-f001:**
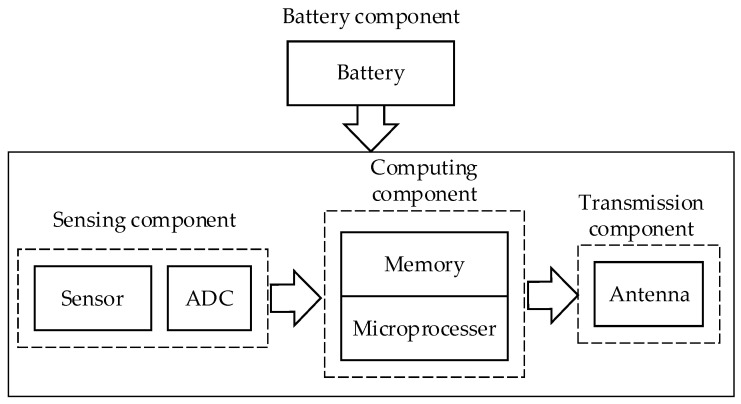
A typical sensor node architecture.

**Figure 2 sensors-23-03846-f002:**
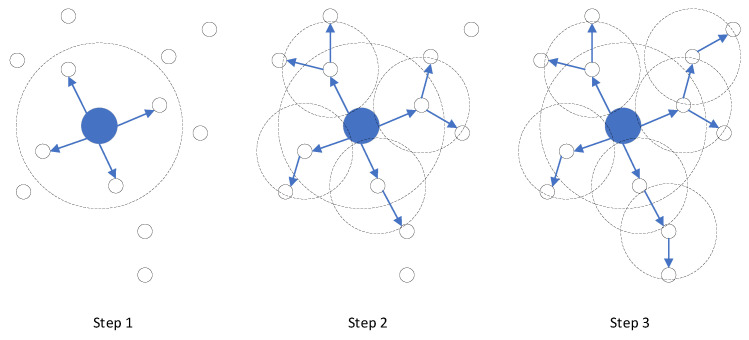
Depth broadcasting process.

**Figure 3 sensors-23-03846-f003:**
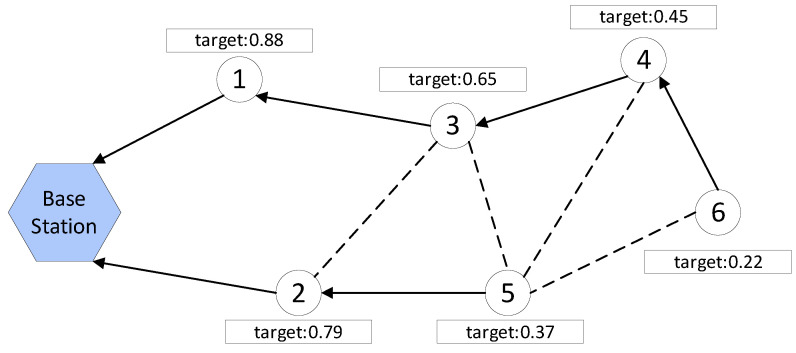
Path discovery example.

**Figure 4 sensors-23-03846-f004:**
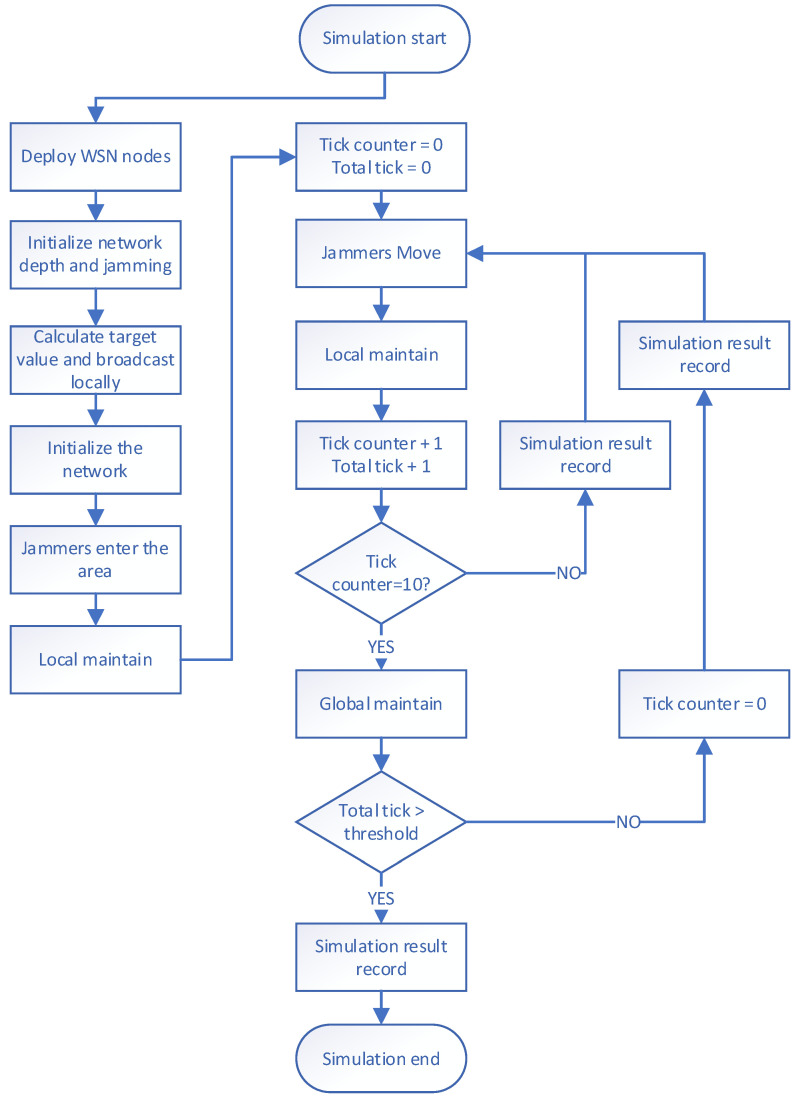
Simulation process.

**Figure 5 sensors-23-03846-f005:**
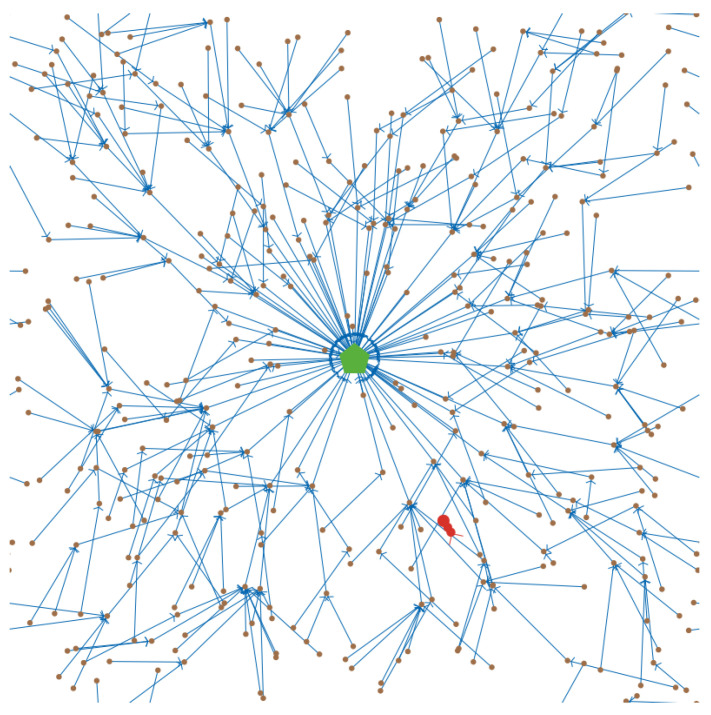
Network instance.

**Figure 6 sensors-23-03846-f006:**
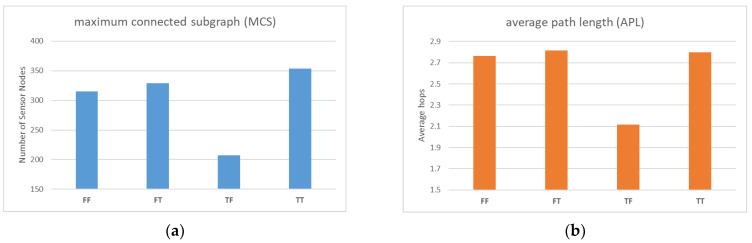
(**a**) Comparison of MCS with different parameter settings; (**b**) Comparison of APL with different parameter settings.

**Figure 7 sensors-23-03846-f007:**
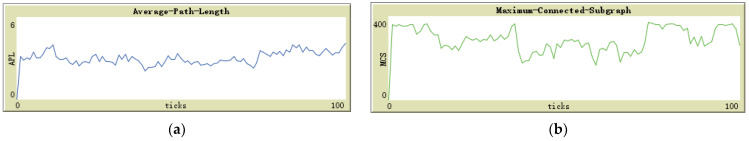
(**a**) APL trend without global maintenance; (**b**) MCS trend without global maintenance.

**Figure 8 sensors-23-03846-f008:**
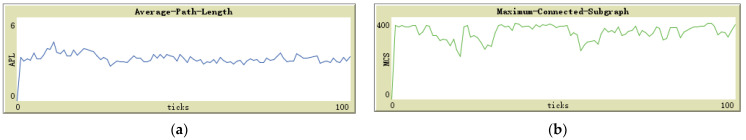
(**a**) APL trend with global maintenance; (**b**) MCS trend with global maintenance.

**Figure 9 sensors-23-03846-f009:**
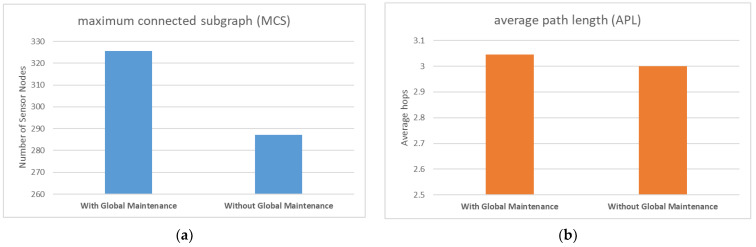
(**a**) Comparison of MCS with or without a global maintenance mechanism; (**b**) Comparison of APL with or without a global maintenance mechanism.

**Figure 10 sensors-23-03846-f010:**
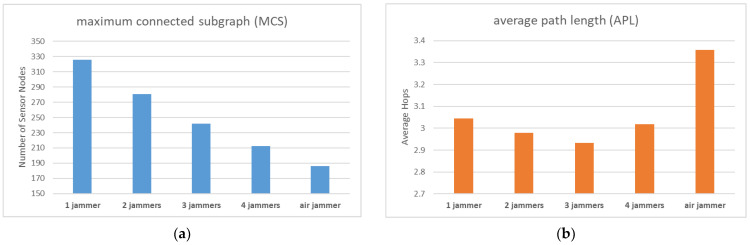
(**a**) MCS comparison of different numbers and locations of jammers; (**b**) APL comparison of different numbers and locations of jammers.

**Figure 11 sensors-23-03846-f011:**
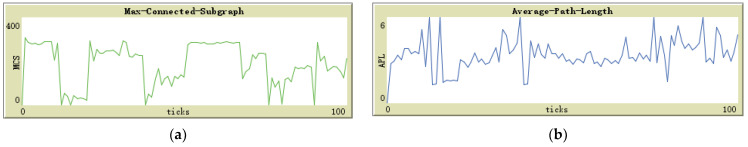
(**a**) MCS trend with air jammer; (**b**) APL trend with air jammer.

**Figure 12 sensors-23-03846-f012:**
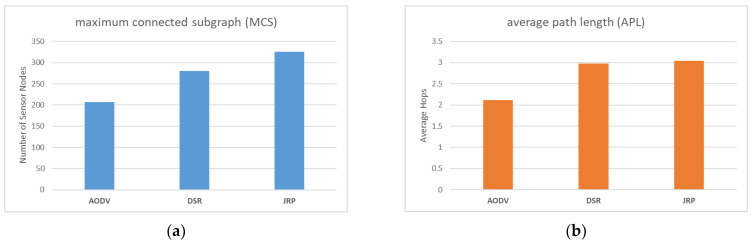
(**a**) Comparison of MCS with different protocols; (**b**) Comparison of APL with different protocols.

**Table 1 sensors-23-03846-t001:** Parameter settings.

Parameters	Simulation 1	Simulation 2	Simulation 3	Simulation 4
Number of sensors	400	400	400	400
Number of base stations	1	1	1	1
Number of Jammers	1	1	{1,2,3,4}	1
Sensor communication distance	8	8	8	8
Base station communication distance	15	15	15	15
Jammer movable	False/True	True	True	True
Jammer Interference Distance—High	16	16	16	24
Jammer Interference Distance—Low	11	11	11	18
Local Maintain	False/True	True	True	True
Global Maintain	False	False/True	True	True
Global Maintenance Frequency		10	10	10
Simulation Duration	1/5	100	100	100

## Data Availability

The simulation files can be found at https://github.com/pauxavi/cyber-physical-social-system-sims (accessed on 1 March 2023).

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
