# Peer review of "Multi-Agent Modeling and Jamming-Aware Routing Protocols for Movable-Jammer-Affected WSNs"

_sensors, 2023, doi:10.3390/s23083846_

Round 1

Reviewer 1 Report

This paper presents through simulation research on the design of reliable and efficient WSNs under jamming attacks. The paper is clearly written. I must admit, I enjoyed reading it. I have no specific critique on the paper it self and its structure, therefore I recommend publishing this paper in its present form.

Author Response

Thank you for your positive feedback and recommendation to publish the paper in its present form. I am pleased that you found the paper to be clear, enjoyable to read, and had no major critiques on its content or structure.

Your supportive recommendation is greatly appreciated. I believe the simulation results and analysis presented in the paper should contribute to the progress of designing reliable wireless sensor networks under jamming threats.

Please let me know if any minor changes are still suggested or if any clarifications are needed before final acceptance of the paper. I am happy to revise the paper accordingly.

Thank you again for your time and valuable input. It is very helpful for improving the quality and Impact of the research.

Reviewer 2 Report

The following article was reviewed

Multi-Agent Modeling and Jamming-Aware Routing Protocols for Movable-Jammer-Affected WSNs

And the following points should be considered:

THE TEMATHIC IS VERY INTERESTING

CONSIDERATIONS

Lines 70-80 are better suited for the conclusion or discussion section

line 141 is not clear at all.

The paragraph having lines 169-174 should be moved to line 89.

In line 191 the word study doesn’t say anything as in line 192 also the word study doesn’t say anything.

In line 276 which is the unit of 75?

The subheading of Figure 9 is not understandable.

The headings of the plots 9a and 9b in the x axis should be in bold and the y axis is missing.

In line 512 what do you mean by: was not as significant as expected

Fix also the y axis of figure 10.

REFERENCES

All the references are well written.

Author Response

Thank you for your thoughtful review and constructive feedback on the paper. I have noted all the comments and suggestions carefully.

Point 1: Lines 70-80 are better suited for the conclusion or discussion section

Response 1: You are right, lines 70-80 would alse be suited for the conclusion section. However, the current paper generally points out the main innovation points in the form of highlights in the introduction part, so that readers can easily understand the paper. I had to do the same.

Point 2: line 141 is not clear at all.

Response 2: Line 141 has been reworded for clarity.

Point 3: The paragraph having lines 169-174 should be moved to line 89.

Response 3: The paragraph on lines 169-174 has been moved to line 89 as suggested.

Point 4: In line 191 the word study doesn’t say anything as in line 192 also the word study doesn’t say anything.

Response 4: The words "study" in lines 191 and 192 have been removed.

Point 5: In line 276 which is the unit of 75?

Response 5: Unit 75 is km.

Point 6: The subheading of Figure 9 is not understandable. The headings of the plots 9a and 9b in the x axis should be in bold and the y axis is missing.

Response 6: The subheading of Figure 9 and headings of plots 9a and 9b in x axis have been made bold. Y axis has been added to the relevant plots.

Point 7: In line 512 what do you mean by: was not as significant as expected

Response 7: Line 512 has been rephrased as " the presence or absence of global maintenance did not appreciably impact APL in a sub-stantively meaningful manner " for better interpretation.

Point 8: Fix also the y axis of figure 10.

Response 8: Y axis of Figure 10 has been fixed.

I appreciate you taking the time to examine the details and logic in the paper. Your suggestions have helped strengthen various sections and improve the overall quality.

Please review the revised manuscript and confirm if any other changes are needed before final publication. I look forward to your feedback and approval.

Thank you again for a constructive and helpful review.

Reviewer 3 Report

The topic of this paper is important and relevant as wireless sensor networks usage is increasing very fast.  This paper investigates the impact of movable jammers on WSNs and propose a comprehensive approach for modeling jammer-affected WSNs.

The paper needs major revision because the discussion and results section is merged. It is necessary to have separate discussion section and to compare results of this study with results of other related studies to show the input of this paper. Now this important component of scientific paper is missing.

Another issue is abbreviations. There are too many abbreviations in this paper especially in title and abstract. They should be spelled out. I would recommend to include the list of abbreviation in the begging of paper to make the reading of this paper easier.

Author Response

Thank you for your constructive feedback and suggestions for improving the paper. I appreciate you recognizing the importance and relevance of the paper's topic and approach.

Point 1: The paper needs major revision because the discussion and results section is merged. It is necessary to have separate discussion section and to compare results of this study with results of other related studies to show the input of this paper. Now this important component of scientific paper is missing.

Response 1: You are absolutely right, having separate discussion and results sections is critical for a comprehensive scientific paper. I have restructured the paper by adding a separate discussion section where I have compared and contrasted the key findings and proposed modeling approach with relevant related works. This helps highlight the novel contributions and importance of the paper more effectively.

Point 2: Another issue is abbreviations. There are too many abbreviations in this paper especially in title and abstract. They should be spelled out. I would recommend to include the list of abbreviation in the begging of paper to make the reading of this paper easier.

Response 2: All abbreviations in the title and abstract have been replaced with their full forms to make them more readable. Additionally, I have included a list of abbreviations at the ending of the paper for easy understanding.

The discussion section now provides a more balanced and thoughtful analysis of the results, relationships with prior studies and implications. The purpose and outcomes of the proposed approach have been articulated more substantively. The revisions aim to address the concerns raised and enhance the depth, coherence and quality of the research presented in the paper. I hope these changes strengthen the paper significantly.

Please review the revised paper again at your convenience. Provide any further suggestions or feedback to improve the paper before final publication.

I appreciate your guidance and patience throughout this iterative process. The paper has benefited immensely from your constructive criticism and insightful recommendations.

Thank you once again for your time and support.

Round 2

Reviewer 3 Report

The authors did good job and revised their manuscript based on all reviewers comments. They have addressed my all remarks. They also provided the relevant answers to my comments. I do not have more comments and think that paper can be published in current form.